# Developing and piloting an online course on osteoporosis using a multidisciplinary multi-institute approach- a cross-sectional qualitative study

**Lena Jafri**[1]*, **Hafsa Majid**[1], **Arsala Jameel Farooqui**[1], **Sibtain Ahmed**[1], **Muhammad Umer Naeem Effendi**[1], **Maseeh-uz Zaman**[2], **Qamar Riaz**[3], **Noreen Nasir**[4], **Sadia Fatima**[5], **Sarah Nadeem**[4], **Rizwan Haroon Rashid**[6], **Aamir Ejaz**[7], **Nusrat Alvi**[8], **Farheen Aslam**[9], **Aysha Habib Khan**[1]

1 Section of Chemical Pathology, Department of Pathology and Laboratory Medicine, Aga Khan University, Karachi, Pakistan, 2 Department of Radiology, Aga Khan University, Karachi, Pakistan, 3 Department of Education Aga Khan University, Karachi, Pakistan, 4 Department of Medicine, Aga Khan University, Karachi, Pakistan, 5 Department of Biological & Biomedical Sciences Aga Khan University, Karachi, Pakistan, 6 Department of Surgery, Aga Khan University, Karachi, Pakistan, 7 Mohi Uddin Islamic Medical College, Mirpur Azad Jamu and Kashmir, Pakistan, 8 Rahbar Medical and Dental College, Lahore, Pakistan, 9 Quaid e Azam Medical College, Bahawalpur, Pakistan

* lena.jafri@aku.edu

**Data Availability Statement:** The datasets generated and/or analyzed during the current study

## Abstract

### Introduction

Postgraduate medical trainees (PGs) in developing nations face various educational hurdles due to limited access to quality resources and training facilities. This study aimed to assess the effectiveness of e-learning, particularly Massive Open Online Courses (MOOCs), within postgraduate medical education. It involved the development of a customized online course focused on osteoporosis for PGs and an examination of their perspectives and preferences concerning online learning methods like Virtual Learning Environment (VLE) platforms.

### Methods

The study was conducted from January 2018 to December 2020. A multi-institutional, multidisciplinary team was assembled to design an osteoporosis course on the VLE platform. PGs (n = 9) from diverse disciplines and institutions were selected with informed consent. Focus group discussions (FGDs) among these PGs identified their preferences for the online course, which subsequently guided the development of the MOOC. The modular MOOC comprised recorded micro-lectures, flashcards, videos, case challenges, and expert interviews. The educational impact of the VLE was assessed using pre- and post-module tests among the participants, and their perceptions of the PGs and course facilitators were gathered via an online survey.

### Results

The study identified the involvement of PGs in the course design process as beneficial, as it allowed for content customization and boosted their motivation for peer-to-peer learning.

are available as coded data without any identifiers on: https://zenodo.org/record/8325306.

**Funding:** The authors received no specific funding for this work.

**Competing interests:** The authors have declared that no competing interests exist.

**Abbreviations:** AKU, Aga Khan University; BMD, Bone Mineral Density; CPSP, College of Physicians and Surgeons Pakistan's; DXA, Dual X-ray Absorptiometry; EMQs, Extended matching questions; FGDs, Focus group discussions; FLS, Fracture Liaison Service; FRAX, Fracture Risk Assessment Tool; MBDs, Metabolic Bone Diseases; MCQs, multiple choice questions; MOOCs, Massive Open Online Courses; OPGs, orthopantomograms; PGs, Postgraduate medical trainees; VLE, Virtual Learning Environment.

During the FGDs, PGs expressed a strong preference for flexible learning formats, particularly short downloadable presentations, and micro-lectures. They also identified challenges related to technology, institutional support, and internet connectivity. In the subsequently customized MOOC course, 66% of PGs (n = 6) attempted the pre-test, achieving a mean score of 43.8%. Following the VLE module, all PGs (n = 9) successfully passed the end-of-module test, averaging a score of 96%, highlighting its impact on learning. The majority (n = 8, 88.9%) agreed that the course content could be applied in clinical practice, and 66.7% (n = 6) expressed extreme satisfaction with the learning objectives and content. Participants favoured end-of-module assessments and the use of best-choice questions for evaluation.

## Conclusion

This study highlights the importance of virtual learning, particularly MOOCs, in addressing the educational challenges faced by developing nations. It emphasizes the need for tailored online courses that cater to the preferences and requirements of PGs. The findings suggest that MOOCs can foster collaboration, networking, and opportunities for professional development, and interdisciplinary collaboration among faculty members can be a key strength in course development. This research provides valuable insights for educators, institutions, and e-learning developers seeking to enhance their teaching methodologies and establish accessible educational environments in the digital age.

## Introduction

As Pakistan's population continues to grow, there is a corresponding increase in the demand for skilled physicians. According to data from 2019, Pakistan had a physician-to-population ratio of 1.1 physicians per 1000 people, which is a staggeringly low number [1]. Life expectancy in Pakistan has expanded from 46.6 years in 1960 to 65.4 years in 2010 and is anticipated to arrive at 72.4 years in 2023 [2]. Population aging and rising rates of osteoporosis, vitamin D deficiency, decreased calcium intake and sedentary lifestyles in Pakistan are some of the rising concerns related to bone health [3–6]. It is estimated that the cost of hip fractures in Pakistan ranges from 1000–10,000 USD, depending on the hospital setting, with an average hospital stay of 7–10 days [7, 8]. Metabolic bone diseases (MBDs) are an ignored clinical entity, and virtual learning environment (VLE) teaching can address training challenges in developing countries by supporting efficient content delivery, reducing physical resources, and increasing access.

A large majority of the educational structures in Pakistan rely on face-to-face teaching. Conventional education patterns of face-to-face teaching or organized projects using printed media cannot meet the adapting needs of healthcare professionals working over a various range of healthcare settings [8]. Postgraduate medical education in Pakistan follows a structured framework with residency training, defined curricula, and assessments. Doctors gain practical experience through clinical rotations and engage in continuous learning through workshops, seminars, and research. The introduction of VLE represents a transformative concept in this field. These technology-enabled platforms overcome geographical barriers, foster interactivity, and collaboration, support continuous professional development, and adapt to changing circumstances [9]. The utilization of web-based learning has developed over the previous decade and offers an adaptable and cost-productive technique to provide education to enormous amounts of learners [10, 11].

Blended learning consistently demonstrates superior effects on knowledge outcomes compared to traditional learning in health education. It provides enhanced access to learning resources, individualized learning experiences, and opportunities for active engagement. These factors contribute to a deeper understanding and retention of knowledge among students [12]. Online learning has the potential to improve pre-clinical medical education by fostering integration and collaboration among different cohorts of medical students, various healthcare professionals, and even across different medical schools, all while preserving elements of social interaction [13]. According to a scoping review by Tang et al., the integration of online lectures into undergraduate medical education seemed to enhance learning outcomes and was particularly welcomed by students [14]. According to a 2018 study by Zhao et al., it is vital for universities to invest in online education and promote the development of Massive Open Online Courses (MOOCs), because they will probably have an inevitable advantage over traditional teaching methods in postgraduate medical education in the near future [15].

Teachers who grew up in a world with limited technology may struggle to use technology to engage and enhance learning. In Pakistan, face-to-face teaching is prevalent, and addressing technological sophistication and introducing virtual learning environments is crucial. The study's objective was to utilize metabolic bone disorders, as a case study to develop MOOC for postgraduate medical students in Pakistan. It aimed to incorporate advanced critical thinking activities, pilot the course with feedback and evaluation, and set it as a model for creating similar courses in diverse healthcare sub-specialties through a VLE. Through this pilot study we aimed to highlight the efforts put in to develop and execute an online course, its educational impact and the evaluation provided by participants and facilitators.

## Materials and methods

A descriptive multi method cross-sectional study was conducted in the section of Chemical Pathology, Department of Pathology and Laboratory Medicine, Aga Khan University (AKU), Karachi, Pakistan in collaboration with AKU's Departments of Radiology, Education, Medicine, Biological & Biomedical Sciences, Surgery along with Mohi Uddin Islamic Medical College, Mirpur Azad Jamu and Kashmir, Pakistan, Rahbar Medical and Dental College, Lahore, Pakistan and Quaid e Azam Medical College, Bahawalpur, Pakistan. The study was conducted from January 2018 to December 2020. The AKU's ethical review committee's permission (ERC number 5415-Pat-ERC-18) was taken, and faculty and PGs provided informed consent before the pilot was conducted.

### Ethics statement

Ethical review processes were followed according to the institutional policies of research practices, which involved obtaining necessary approvals and adhering to stipulated guidelines and regulations. Participants who consented to be in the study were treated with respect, and their privacy and confidentiality were prioritized. Participant identities were anonymized upon data entry and these codes were referred to throughout the study. Digital data was secured on a computer with an encrypted server belonging to the PI, and only authorized team personnel had access to the data upon taking permission from the PI. As per institutional policies, data is retained for a period of seven years, after which it will be deleted. Authors have maintained transparency and disclosure in reporting findings and disclosing conflicts of interest.

### Data collection

The course director created a multidisciplinary facilitator team that included faculty from various institutes. The College of Physicians and Surgeons Pakistan's (CPSP) registered

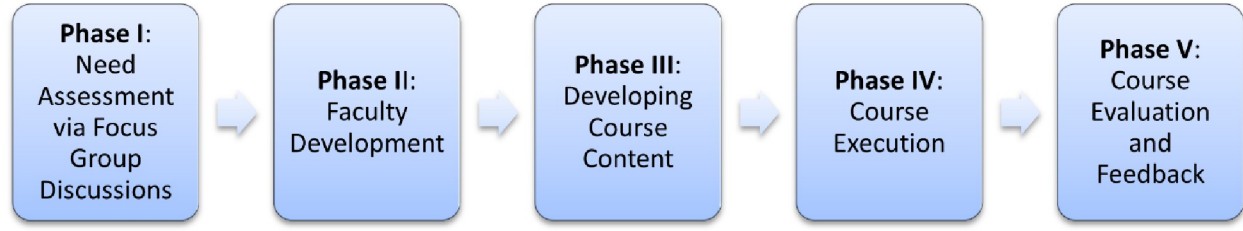

**Fig 1. Methodological workflow of the study.**

postgraduate residents (PG) from various specialties were invited to participate through simple random sampling. Convenience sampling was done to identify facilitators with experience and expertise in the subject area (osteoporosis) of the online course. To ensure a comprehensive analysis and gather diverse perspectives, two separate groups of PGs were involved in this study. The first group of PGs participated in the Focus Group Discussions (FGDs) and second group in the course execution. To make sure the results of each stage meet their mission, and that the course module team is ready to smoothly transition to the next phase, we developed the workflow in the following five phases, outlined in Fig 1.

## Phase I—Need assessment via FGDs

The project team felt the need to have the PGs' perception of an MOOC on osteoporosis before developing the course content. Before initiating the discussion, written informed consent was taken from each PG trainee. Consent was taken via emails from PGs residing in other cities and it was clearly communicated that they were being invited to participate in this study as learners only and had the right to refuse. They were informed that there were no right or wrong answers to the questions to be asked and were expected to have differing points of view. The entire process of FGDs is summarized in Fig 2.

## Phase II—Faculty development

A multi-disciplinary, multi-institute facilitator team was formulated with people who shared a common interest in designing an online course on MBDs. A complete hands-on workshop was conducted on Moodle by the blended and digital learning network of the institute with facilitators from other cities connected via online platform Zoom.

## Phase III—Developing course content

The course blueprint and course content were designed keeping in view the needs of PG trainees and challenges highlighted in previously conducted FGDs. The team tried minimizing required reading and relevant questions were developed to assess the trainees. The course outlines were developed by the faculty to establish a connection between the pathophysiology of various MBDs and their respective risk factors and signs and symptoms. Multiple disciplines contributed to development of course content, leveraging their knowledge to improve the learning experience (Tables 2 and 3). Special focus was laid on teaching participants how to order appropriate investigations to make a diagnosis of MBDs and prescribe pharmacologic and non-pharmacological interventions for the prevention and treatment of MBDs.

The course content included a series of required readings, lectures, videos, flashcards, case challenges, and discussion forums. Screengrabs of the VLE course platform are shown in Fig 3a–3c.

## Phase IV—Course execution

After blueprinting Moodle with the course outline, the facilitators (Four faculty members and a research associate from Chemical Pathology section, AKU) identified PGs from different disciplines and various institutes across Pakistan, who were then enrolled on Moodle to take up the course at their convenience. Email communications and e-meetings were carried out to keep the team updated and active team members were acknowledged timely. Announcements

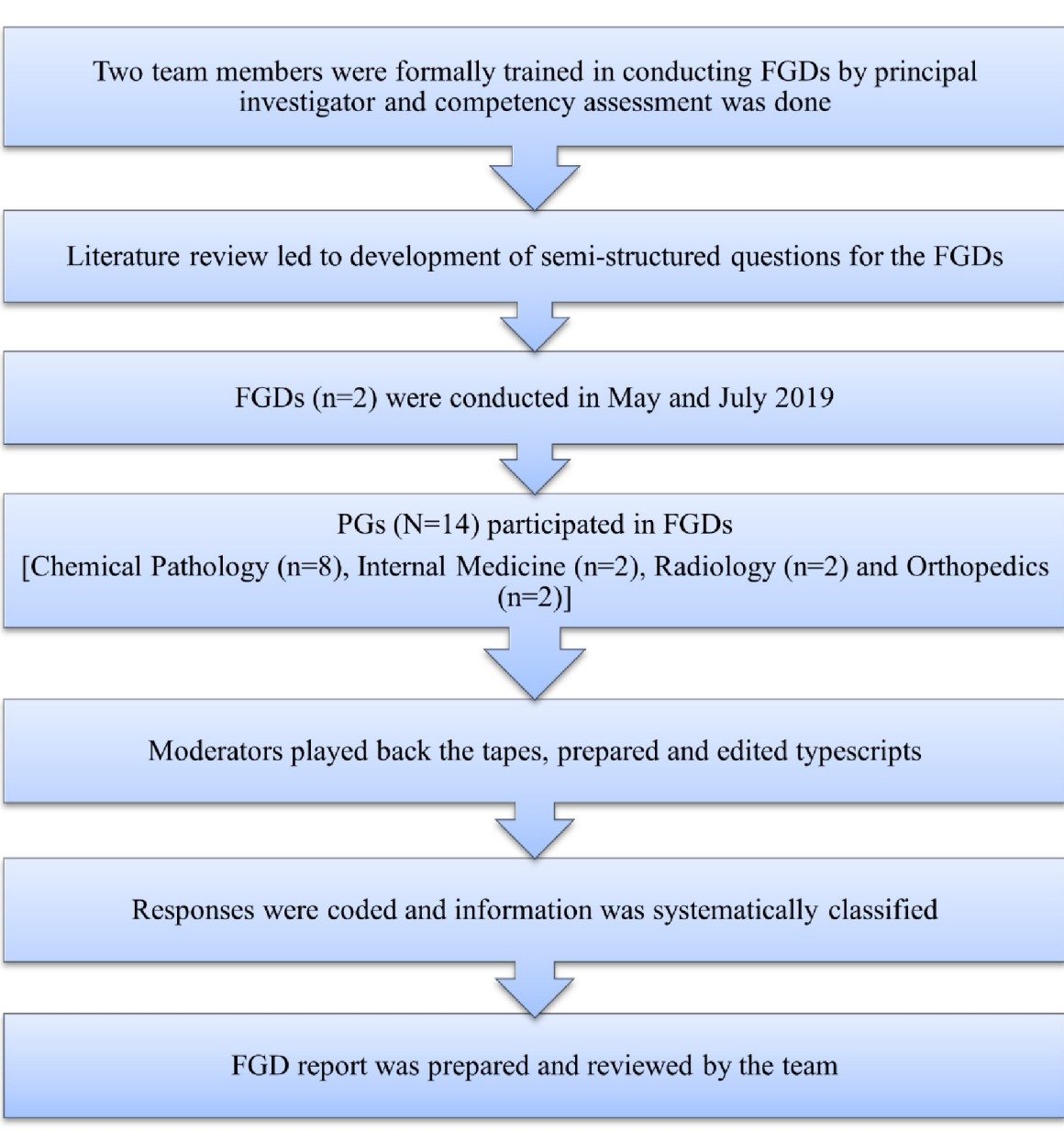

**Fig 2. Flowchart outlining the process of focus group discussions.**

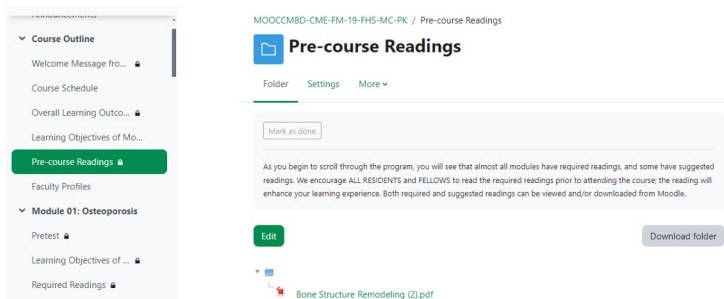

**Fig 3.** a: Example of Pre-course Readings Required for Undertaking the VLE Course. b: Example of Diagnostic Case Challenge Included in the VLE Course. c: Risk Factors Associated with the Diagnostic Case Challenge.

were made on the VLE platform every time an open discussion on a particular topic was planned. A MOOC WhatsApp group was also created to keep participants and active team members in the loop and address the PGs' issues when required.

## Phase V—Course evaluation and feedback

Before the required reading for the module and other education material were made available, PGs were made to take the pre-module test which consisted of scenario-based multiple-choice questions. After the module was completed, the participants had to appear in the end-of-module quiz to gauge the gain in their knowledge. The post-test consisted of eight Best Choice Questions (BCQs) covering the learning objectives of the osteoporosis module. The levels were set as follows:

- Pretest

- Formative: Extended matching questions (EMQs), cases. Feedback will be given to those below the level, i.e. those unable to pass the course.

- Summative (to determine terminal performance) or post-test:

  - Selection of type of test paper: BCQs, Real world problems or cases will be used. Level of difficulty in BCQs and EMQs will be C2 and C3 only.

  - Each module and end of course summative exam need to be passed. Passing score would be 65%.

  - Those who fail will be given some time to revisit the teaching material and re-attempt the quiz until they pass. (time lapse, bank, 1/2 supplementary attempts allowed)

Besides this, an online survey was performed to gather feedback from PGs and facilitators regarding their experience of the pilot MOOC in terms of course content and satisfaction of this innovative teaching and learning method.

## Statistical analysis

The qualitative analytical method proposed by Braun and Clarke was used to conduct thematic analysis of the FGD following the six phases [16]. The six steps of analysis included data familiarization, generating initial codes, identifying themes, reviewing themes, defining themes, and report writing. Moderators played back the tapes, prepared and edited typescripts,

performed coding, and systemically classified the information. Data collected was checked for accuracy by the principal investigator, and all team members reviewed the FGD report.

For quantitative data of FGDs, and PGs and facilitators perspective about the course, data was analyzed by Microsoft Excel. Frequency and percentages were generated.

## Results

### Phase I—Need assessment via FGDs

In total, two FGDs were conducted which lasted for one and a half hours to two hours each. Each FGD consisted of 5–7 participants. Altogether fourteen PG trainees participated in the two FGDs, who were from all levels of residency training (Years 1–6). The male to female ratio was 2:1. Residents from AKU-Karachi took part in the discussion physically, while residents from Rehman Medical Centre-Peshawar, and Quaid-e-Azam University-Bahawalpur, participated remotely via Zoom. Four main themes emerged from analysis of PGs discussions in FGDs: need of course, time commitments, tools and technology and course content/assessment, which are summarized in Table 1.

### Phase II—Faculty development

A multi-disciplinary, multi-institute facilitator team (n = 17) was formulated with experts from Radiology, Orthopedics, Medicine, Endocrinology, Chemical Pathology and Education, from across Pakistan. The institution's blended and digital learning network trained the facilitators (n = 7) on how to adapt themselves to virtual learning environment (Moodle). The sessions presented an overview of the VLE platform, highlighting its features, functionalities, and potential teaching and learning benefits. Practical training sessions gave facilitators the opportunity to explore the platform in a safe setting, allowing them to experiment and become acquainted with its numerous tools and resources. Research assistant and associates from the department of Pathology and Laboratory Medicine, AKU, were also trained in this workshop. The entire process led to development and strengthening of skills, instinct, and abilities that organizations and communities need to survive, adapt, and thrive in a fast-changing world.

**Table 1. Results of the focus group discussion with postgraduate residents (n = 14) on needs of metabolic bone disorders (MBDs) online course in their training.**

| Themes | Findings of Focus Group Discussion -PGs responses | n (%) |
|---|---|---|
| The need for Metabolic Bone Disorders -course on Virtual Learning Environment | There is an urgent need for a course on MBDs | 14 (100) |
| | The burden of MBDs is high in Pakistan | 8 (57.1) |
| | Current residency training does not appropriately cover objectives related to MBDs | 10 (71.4) |
| Time Commitment | Weekdays-early evenings are the preferred time to access the MBDs course | 9 (64.2) |
| | Total time that can be taken out from residents' schedule is 1–2 hours per week | 9 (64.2) |
| Tools and Technology use | Course access via laptop is preferred over cellphone | 9 (64.2) |
| | Comfortable in using social media group as a tool for connecting with course faculty | 12 (85.7) |
| | Internet connectivity and bandwidth may become a barrier in course access | 10 (71.4) |
| Course content and assessment | Such courses on VLE should include short or micro lectures and/or downloadable presentations | 12 (85.7) |
| | Best choice questions should be used for end of module assessments | 14 (100) |

These trained facilitators further managed Moodle and trained additional faculty members from Mohi Uddin Islamic Medical College, Mirpur Azad Jamu and Kashmir, Pakistan, Rahbar Medical and Dental College, Lahore, Pakistan and Quaid e Azam Medical College, Bahawalpur, Pakistan on the use of Moodle for teaching. The facilitators who were taught in developing and running a VLE course trained more facilitators, creating a ripple effect.

## Phase III—Developing course content

The team developed a blueprint for the four modules of the course. Four modules were decided: osteoporosis, osteomalacia and rickets, parathyroid disorders, and rare bone diseases. Osteoporosis was chosen as the pilot module and its complete course content was uploaded on Moodle, outlined in Tables 2 and 3.

The faculty developed the content based mostly on case scenarios posing critical thinking and decision making. The teaching tools used as the course guide for the facilitators are shown in Table 4. Challenges and fears of PGs from FGDs were critically analysed by the project team and facilitators, who then produced solutions to the problems, which are summarized in Table 5. Questions developed for assessment at the end of the module were reviewed by educational experts and content related evidence was reviewed by five other facilitators for validity and feasibility.

## Phase IV—Course execution

The pilot for the virtual osteoporosis module was spread over a duration of one month (June–July 2019). Nine PG trainees were enrolled in the course and included fellows and residents' years 1–5 from Chemical Pathology (n = 6), Endocrinology (n = 1), Radiology (n = 1) and

**Table 2. Material developed for osteoporosis module on virtual learning environment by multidisciplinary facilitator team.**

| Type of content | Description | Contact Time |
|---|---|---|
| **Flash cards** | • Bone turnover markers<br>• Fracture Risk Assessment Tool (FRAX) limitations<br>• Fracture Liaison Services | • 10 mins<br>• 10 mins<br>• 10 mins |
| **Videos** | • Overview of Osteoporosis<br>• Secondary Osteoporosis<br>• DXA-scan procedure and interpretation<br>• Expert's insight on Osteoporosis treatment | • 8 mins<br>• 10 mins<br>• 6 mins 30 secs<br>• 4 mins 15 secs |
| **Documents** | • Course curriculum<br>• Overall learning outcomes<br>• Learning objectives of module 1<br>• Faculty profiles | • 5 mins<br>• 5 mins<br>• 10 mins |
| **Educational content** | • Pre-course reading on bone structure and remodeling by senior faculty<br>• Review article | • 10 mins<br>• 20 mins |
| **Pre-test, end of module quiz and Case challenges** | • BCQs based pre-test and end of module quiz<br>• Scenario based case challenges on osteoporosis pathophysiology, diagnosis and management | • 40 mins<br>• 45 mins |

Abbreviations: FRAX (fracture risk assessment score), mins (minutes), secs (seconds), BCQs (best choice questions)

**Table 3. Learning objectives for the osteoporosis module.**

| Course Objectives |
| --- |
| • Understand the prevalence of osteoporosis by age and gender in Pakistan |
| • Define fragility fractures and its burden in Pakistan |
| • Identify most common and least likely sites of fragility fractures |
| • Diagnose fragility fractures based on relevant biochemical investigations |
| • Suggest measures to reduce recurrent fractures |
| • Understand the role of nutrition and lifestyle factors in prevention of osteoporosis |
| • Recommend dietary calcium intake for patients with osteoporosis |
| • Know the risk factors included in FRAX |
| • Apply fracture risk assessment tool on patients |
| • Appreciate the limitations of FRAX in Pakistani population |
| • List different bone turnover markers and describe its clinical utility and limitations |
| • List indication for performing X-rays, Dual X-ray Absorptiometry (DXA) scans in patients with osteoporosis |
| • Order appropriate and cost-effective investigations including X-rays, DXA scans (ensuring differentiation Z-score and T-score) |
| • Interpretation DXA and make a diagnosis of osteoporosis and osteopenia based on DXA report |
| • List the indications for ordering a vertebral fracture assessment and its limitations |
| • Knowledge of other available methods for Bone Mineral Density (BMD) measurement |
| • Know the mechanism of action, use and rare side effects of following drugs:<br> • Oral bisphosphonates–alendronate, risedronate and ibandronate<br> • IV bisphosphonates–zoledronate, ibandronic acid<br> • Strontium ranelate<br> • Denosumab<br> • Raloxifene<br> • Teriparatide<br> • Hormone Replacement Therapy (with/without progesterone)<br> • Calcium<br> • Vitamin D |
| • Prescribe pharmacologic and non-pharmacological interventions for the prevention and treatment of osteoporosis |
| • Understand role of a multidisciplinary group in management of osteoporotic fracture |
| • List the various components in a Fracture Liaison Service (FLS) |
| • Develop liaison for FLS and the pathway for its implementation |

Internal Medicine (n = 1) belonging to institutes of Aga Khan University Karachi, Chughtai Institute of Pathology, Lahore, Rehman Medical Institute, Peshawar and Quaid-e-Azam Medical College, Bahawalpur. The PGs attempted the pre-test and completed the osteoporosis module. Three attempts were allowed for the end of module assessment, two trainees performed it

**Table 4. Results of focus group discussion with metabolic bone disorders online course facilitators on its designing and execution.**

| Challenges identified | Solutions proposed |
| --- | --- |
| Need for e-course | Pedagogical model prepared |
| Short attention spans of PG trainees | Prepared Micro lectures (max 7 min) |
| Interactive Course Content | Teaching and learning activities in the form of case challenges and discussion forum |
| Internet connectivity issues | Downloadable lectures |
| Busy schedule of trainees | Course access flexibility |
| Limited learning time available to trainees | Kept it asynchronous with division into multiple modules |

**Table 5. Teaching tools used for the osteoporosis module.**

| Days | Activity | Accessibility | Student engagement level | Faculty engagement level during the course | Contact Time in min |
|------|----------|---------------|--------------------------|--------------------------------------------|---------------------|
| 3 days before opening the course | Open Welcome Message by Course Director<br>Faculty Profiles with pictures<br>Open Course Outline<br>Ice breaking Activity | Open till end<br>Open till end<br>Open till end | +<br>+<br>+++ | -<br>-<br>+++ | 5<br>5<br>15 |
| 1 | Open Module 1 objectives<br>Open Required Reading for module 1<br>Open Video Lecture on Osteoporosis | Open till end<br>Open till end<br>Open till end | +<br>+<br>++ | -<br>-<br>+ | 10<br>20<br>8 |
| 2 | Case Challenge 1 | Open till end | +++ | - | 15 |
| 3 | Open Video on Sec Osteoporosis<br>Open flash card on bone turnover markers | Open till end<br>Open till end | ++<br>+ | +<br>- | 10<br>10 |
| 4 | Case Challenge 2 | Open till end | +++ | - | 15 |
| 5 | Open both Videos on DXA Scan | Open till end | +++ | - | 5:30<br>1 |
| 6 | Case Challenge 3 | Open till end | +++ | - | 15 |
| 7 | Open Discussion Forum on FRAX | Open for 4 days | +++ | +++ | 15 |
| 8 | Open Flash Card on FRAX limitations | Open till end | + | - | 10 |
| 9 | Case Challenge 4 | Open till end | +++ | - | 30 |
| 10 | Open video on experts' insight into osteoporosis treatment | Open till end | +++ | - | 4:15 |
| 11 | Buffer Time | - | - | - | |
| 12 | Open Flash card on Fracture Liaison Service | Open till end | + | - | 10 |
| 13 | Buffer Time | - | - | - | |
| 14 | End of module test | Open for 2 days | +++ | - | 20 |
| 16 | Course completed; VLE shell hidden from students; certificate provided to those who attempted end of module quiz | Open for 5 days | + | +++ | |

Students contact time: 209 minutes in all = 3.4 hours min to 5 hours max

thrice while the rest attempted it twice. In the end of module assessment eight participants scored 100% while one scored 87.5%. The course completion rate of all nine participants within the time frame of one month was 100%.

The course faculty remained connected during the course's execution to facilitate student interactions and respond to questions on Moodle in the discussion forum as well as on WhatsApp. For notifications, clarifications, and individualized comments, Moodle's messaging system was used. During this execution period, two discussion threads were formed, with nine responses from nine different participants and 14 responses from nine individuals. This phase of execution encouraged facilitators to actively participate in discussions, and share their experiences. Collaboration and peer learning were encouraged, resulting in a supportive community of facilitators who could share ideas and best practices for efficiently using the VLE.

## Phase V—Course evaluation and feedback

Results of the pre-test and end of module quiz are shown in Fig 4. The PGs were invited to submit comments on the course during the evaluation process, taking into consideration their experiences using the VLE platform as a tool for teaching and learning, as well as their perception of the course content. Most of the participants were in strong consensus for this way of teaching and learning and were satisfied with the course content as shown in Fig 5a. PGs

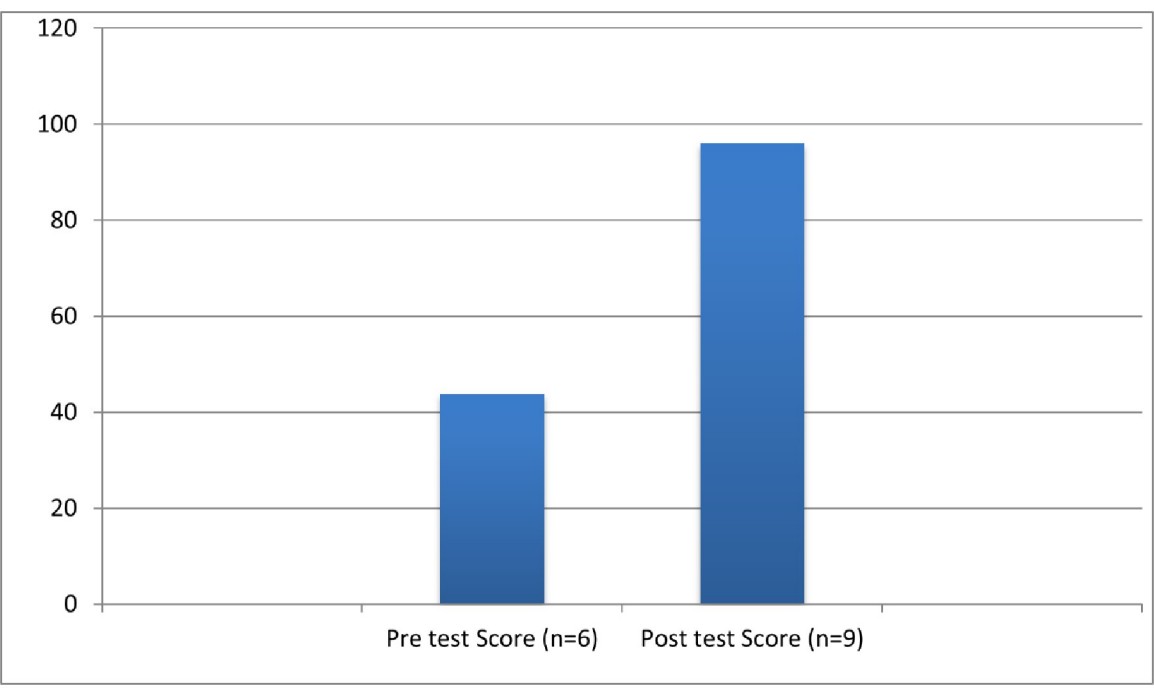

**Fig 4. Pre and post-test scores for the osteoporosis module of postgraduate trainees.**

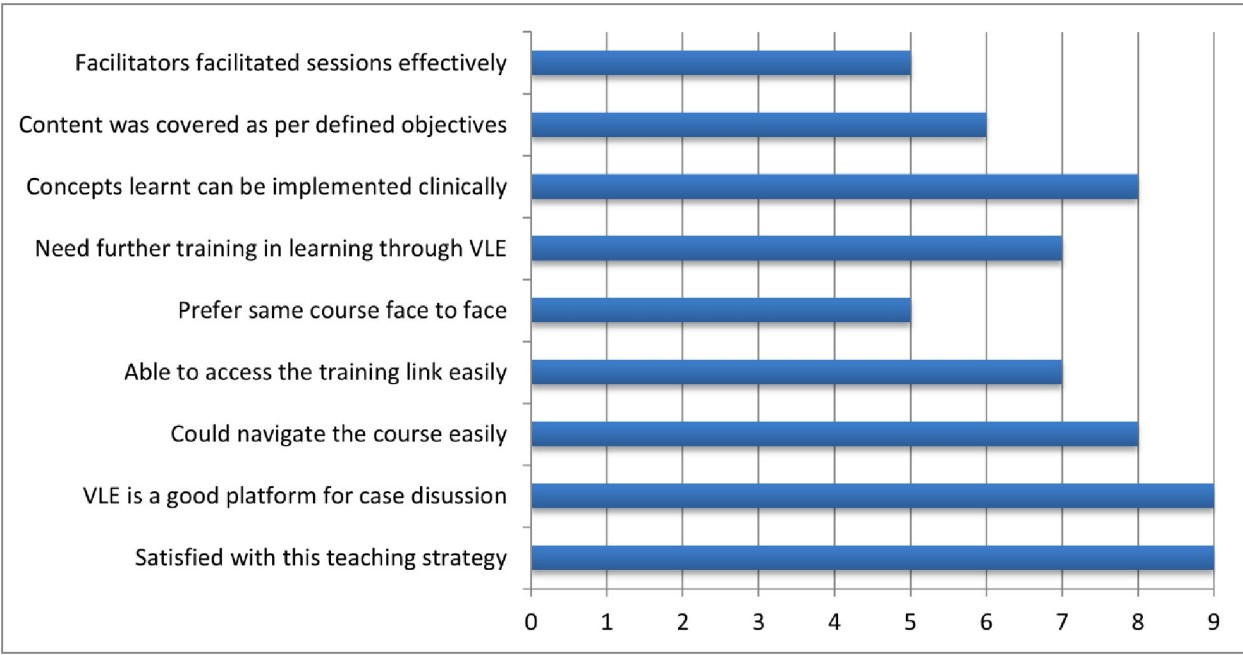

**Fig 5.** a: Feedback of Postgraduate trainees for VLE as a teaching-learning methodology and Course Content (n = 9). b: Feedback of Course Facilitators for VLE as a teaching-learning methodology and on Course Content (n = 8).

acknowledged the efficiency of the VLE platform as a medium for delivering course content based on their own experiences. They felt it to be a wonderful resource that aided in their involvement and learning. The VLE platform's flexibility and accessibility enabled them to access course materials at their leisure, enabling self-paced learning and developing a deeper comprehension of the subject matter. Participants acknowledged their gratitude for the quality and relevancy of the course material in addition to their good feedback about the VLE platform. They thought the content was thorough, well-structured, and relevant to their learning requirements and aspirations. The course material created a firm foundation for their comprehension of pathophysiology of osteoporosis and its management, and participants recognized its importance in assisting them on their learning path.

Around 47% (8/17) of the facilitators participated in an online survey to provide their feedback of virtual teaching experience. All facilitators were extremely content with various aspects of this innovative teaching methodology; however, they shared that they still need further assistance to master the art (Fig 5b). Though feasibility was not discussed as a separate aspect of the course feedback, responses from the facilitators and participants made it evident that on the whole, they were able to navigate the course easily and access it at any time of the day as per their convenience. Participants were encouraged to share their honest feedback regarding the various aspects of the course (I.e. network connectivity, access to technology, bandwidth considerations, localization and language, learner support, capacity building on VLE application, and sustainability), but the investigators did not receive any negative feedback, hence it was concluded that the course was found to be feasible.

## Discussion

E-learning has shown to be one of the tools that addresses some of the training challenges experienced in developing countries by supporting efficient content delivery, decreasing costs, and increasing access [17]. Qualitative analysis of the participating PGs' perspectives in FGDs revealed themes that helped design a completely student-focused online course for osteoporosis.

The utility of involving PGs in course formulation process is highlighted when the course content is being designed, as their opinions bring relevant changes to the content, and enhance their eagerness and motivation for learning, ultimately promoting peer to peer learning and offering a course that is customized according to their needs. The virtual design of the course facilitates professional development for healthcare providers on a bigger scale, bridging those who do not have direct access to the course material in person. Some issues that may occur in designing the course circulate around technological difficulties, familiarizing training faculty with the teaching modalities of the virtual platform, and effective use of online resources by participants and facilitators. Other barriers that hinder the development and adoption of online learning in medical education include limited time availability, a lack of institutional strategies and support, and negative attitudes among the involved stakeholders [18].

In our study, participants and facilitators did not share major negative feedback regarding the VLE platform as users. Current gaps in course content were outlined by the PGs who shared that they were not taught sufficient material about MBDs, and the course content was consequently developed to address topics that would be relevant during their residency exit examinations. According to the trainees, the average time that they would devote to studying the BMD course was two hours per week, hence the duration of the course content was customized accordingly. Moreover, most of them wanted to learn in the afternoon during the weekdays on-campus. A discovery by Green et al. also showed that students on blended courses preferred using on-campus time more, primarily to discuss with other colleagues [19].

With the advent of innovative educational methodologies such as flipped classrooms, live online lectures are being utilized for distant learning programs. However, our focus group had a contrasting opinion, with only two participants in concordance with the idea of using this approach. The group claimed that live lectures require a dedicated time slot, which is challenging to manage with routine duties. The majority of participants were in favor of short downloadable presentations or short duration micro-lectures, which have comprehensive content and would enable PGs to study as per their convenience.

Continuity in learning as an essential aspect of online learning has positively shown to affect overall gain in skills and knowledge in clinical reasoning [20]. The participating PGs also positively responded to the VLE platform, appreciating that it removed their digital inexperience and providing them with a flexible remote learning strategy. Numerous studies performed through questionnaires also showed high acceptance of using computer assisted learning [21, 22]. The findings of this study strengthened the results of previously evaluated studies where students preferred using computer-based educational material [23–25]. It must be acknowledged that although the VLE platform for this course is designed to be user-friendly and is accessible by computers and handheld devices, i.e., smartphones and tablets, some PGs shared concerns about internet connectivity problems being a possible barrier to access.

E-assessment and evaluation of practice are part of the e-learning strategy and are thus essential to ensure its utility. On being asked what point of the course evaluations should be carried out at, the cluster had a diversified opinion, with most in favour of end of module evaluations. This methodology is evidenced to make participants more focused to qualify for the next session, with the provision of scores in each module nurturing a competitive learning environment [26]. On a bigger scale, E-assessment also permits institutions to widen the range of evaluation practices and improves the impartiality and stability of their assessment. It also diminishes the marking burdens and helps the organization and assessment programs and evaluation techniques [27].

The group had a consensus regarding the use of best-choice questions (BCQs) as the mode of assessment. They accentuated that the focus of BCQs should be on diagnostic, case based, data driven approaches to feed their diagnostic capabilities of MBDs. In their view, case-based teaching and assessment provide better judgment of a candidate's understanding of the entire clinical spectrum.

Findings of our study support the importance of faculty development by designing an online course that is effective, flexible, and accessible, after overcoming all challenges and hurdles. Involving multiple disciplines endorses the collaborative integrative model of patient care required for MBDs. By integrating the expertise of these disciplines, the course provided a comprehensive understanding of the pathophysiology, diagnosis, treatment, and management of osteoporosis. Upgrading the aptitudes of faculty members as postgraduate supervisors is imperative to ensure a fruitful supervision experience and to diminish the whittling rates among PG students [28]. On-going research reveals that MOOCs offer a favorable solution for faculty development [29]; associated advantages of utilizing a MOOC with faculty peers may incorporate expanded awareness and comprehension about web-based learning through direct insight.

In this pilot course on osteoporosis, having faculty from multiple disciplines including Radiology, Orthopedics, Medicine, Endocrinology, Chemical Pathology and Education strengthened the quality of course material delivered to the PGs. Prospectively, the course may include faculty from Dentistry, as studies highlight the significance of full-mouth X-rays or orthopantomograms (OPGs) in establishing a diagnosis of osteoporosis. According to recent studies, radiographic findings on OPGs can help the dentist identify patients with low bone density, and then refer them to specialists for BMD assessment [30]. Computer active shape

modelling technology can be used to detect bone density on OPGs by measuring the cortical width of the bone on the lower border of the mandible. This automatic method is reproducible and has good diagnostic accuracy (Az = 0·759 (95% CI = 0·724–0·791) [31]. This finding is supported by the meta-analysis performed by Calciolari et al., which explored three different mandibular bone measurement indices with variable sensitivity and reported the best specificity in measuring the mandibular cortical width, also known as the mandibular cortical thickness, or mental index [32]. Given its abundant citations in literature, the utility of dental OPGs in detecting osteoporotic risk is an area that can be inexpensively explored and incorporated in future curricular courses.

If online curricula are constructed and organized diligently, students across different institutions can easily access open academic lectures at ease. If PGs are specifically targeted and course content is related to their professional needs and research interests, online learning platforms can prove to be greatly beneficial to their academic growth and success [33]. The present study conducted on postgraduate medical education curriculum needs was the first of its kind in Pakistan. The type of study design gave a clearer picture of various PGs' perceptions towards the acceptance of distance learning. Their perspectives helped the project team in designing a successful MBD online course catering to the exposed challenges.

## Conclusion and recommendation

In this digital era, it is imperative that PGs are exposed to learning pedagogical strategies via the VLE. The VLE platform for osteoporosis course has proved to provide PGs with a supportive learning tool that addresses training challenges faced by developing countries, including cost, content delivery and remote accessibility. The subsequent customized MOOC course significantly improved learning outcomes, with high participant satisfaction and applicability to clinical practice. Possible advancements for the future of MOOCs in the context of PG training may include specialized courses, inter university collaborations, integration with blended learning approaches, or developing personalized learning paths for future growth. The teaching-learning environment fostered through these courses may be beneficial for collaboration and networking and provide continued professional development options for the users. Findings of this study will be essential for faculty members, institutions' administrations and e-learning developers who are keen to effectively design MOOCs as a pliable and constructive source for learning. They will be able to design and implement more feedback-based courses to bring innovation to their teaching-learning strategies and promote an accessible environment for education. The incorporation of a VLE into medical education can catalyze a change towards using adult learning theory, in which educators will become more involved as facilitators of learning rather than just material providers.

## Acknowledgments

This project was part of the PI's Foundation for Advancement of International Medical Education and Research (FAIMER) fellowship from Philadelphia, and we would like to acknowledge them. We extend our gratitude to Diane Shoemaker, who served as the project advisor from FAIMER, as well as William Burdick and Ryan Emery from FAIMER for their valuable guidance on microteaching. We would also like to express our appreciation to Nadir Shah and Eman Rashwan from the Department of IT Academics and Computing at AKUH for conducting informative workshops on MOODLE and offering unwavering support to the VLE Champs team.

## Author Contributions

**Conceptualization:** Lena Jafri, Aysha Habib Khan.

**Data curation:** Lena Jafri, Hafsa Majid, Sibtain Ahmed, Muhammad Umer Naeem Effendi, Maseeh-uz Zaman, Qamar Riaz, Noreen Nasir, Sadia Fatima, Sarah Nadeem, Rizwan Haroon Rashid, Aamir Ejaz, Nusrat Alvi, Farheen Aslam, Aysha Habib Khan.

**Formal analysis:** Lena Jafri, Hafsa Majid, Sibtain Ahmed, Muhammad Umer Naeem Effendi.

**Investigation:** Lena Jafri, Hafsa Majid, Sibtain Ahmed.

**Methodology:** Lena Jafri, Hafsa Majid, Sibtain Ahmed, Maseeh-uz Zaman, Qamar Riaz, Noreen Nasir, Sadia Fatima, Sarah Nadeem, Rizwan Haroon Rashid, Aamir Ejaz, Nusrat Alvi, Farheen Aslam, Aysha Habib Khan.

**Project administration:** Lena Jafri, Sibtain Ahmed.

**Resources:** Lena Jafri.

**Software:** Lena Jafri.

**Supervision:** Lena Jafri, Aysha Habib Khan.

**Validation:** Lena Jafri.

**Visualization:** Lena Jafri, Arsala Jameel Farooqui.

**Writing – original draft:** Lena Jafri, Arsala Jameel Farooqui, Muhammad Umer Naeem Effendi, Aysha Habib Khan.

**Writing – review & editing:** Lena Jafri, Hafsa Majid, Arsala Jameel Farooqui, Sibtain Ahmed, Muhammad Umer Naeem Effendi, Aysha Habib Khan.

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
