## [Decision Letter · Decision Letter 0]

12 May 2023

PONE-D-22-28614Developing and piloting an online course on osteoporosis using a multidisciplinary multi-institute approach- a cross-sectional qualitative studyPLOS ONE

Dear Dr. Jafri,

Thank you for submitting your manuscript to PLOS ONE. After careful consideration, we feel that it has merit but does not fully meet PLOS ONE’s publication criteria as it currently stands. Therefore, we invite you to submit a revised version of the manuscript that addresses the points raised during the review process.

We look forward to receiving your revised manuscript.

Kind regards,

Mukhtiar Baig, Ph.D.

Academic Editor

PLOS ONE

Additional Editor Comments:

It needs major revision.

Reviewers' comments:

Reviewer's Responses to Questions

**Comments to the Author**

1. Is the manuscript technically sound, and do the data support the conclusions?

Reviewer #1: Yes

Reviewer #2: Partly

Reviewer #3: Partly

Reviewer #4: Partly

2. Has the statistical analysis been performed appropriately and rigorously? 

Reviewer #1: Yes

Reviewer #2: Yes

Reviewer #3: N/A

Reviewer #4: N/A

3. Have the authors made all data underlying the findings in their manuscript fully available?

Reviewer #1: No

Reviewer #2: No

Reviewer #3: Yes

Reviewer #4: Yes

4. Is the manuscript presented in an intelligible fashion and written in standard English?

Reviewer #1: Yes

Reviewer #2: Yes

Reviewer #3: No

Reviewer #4: Yes

5. Review Comments to the Author

Reviewer #1: Thank you for giving me the opportunity to review this well perceived and well written article titled, "Developing and piloting an online course on osteoporosis using a multidisciplinary multi-institute approach- a cross-sectional qualitative study". The methodology adopted is elaborated in detail and is reproducible. Moreover there is excellent flow of concepts. Only the references that are used are old. The authors need to discuss the results in light of recent literature preferably from within the last 5 years.

Reviewer #2: Paper Review: Developing and piloting an online course on osteoporosis using a multidisciplinary multi-

institute approach- a cross-sectional qualitative study

This paper was about a cross-sectional qualitative study on a pilot online course of osteoporosis using a multidisciplinary multi-institute approach. It should be helpful for the postgraduates (PGs) in the medical education system to have the opportunities to learn from a virtual collaborative environment. This study showed that it was possible to design a multidisciplinary online course on osteoporosis, but the evidence was rather weak in terms of the educational impact of a novel educational tool described in this manuscript. The participating PGs were nominated by their faculty- the sample probably was not representative of the PG population in the country of study. Also, the qualitative nature of this study was not the best approach to examine the educational impact of this ‘novel’ tool. The main research question depicted in the abstract seemed to be ‘what was the educational impact of this novel educational tool on teaching and learning capabilities of PGs and course facilitators?”, and therefore the results only focused on the ‘quantitative’ findings based on the different measures related to the online course. Overall, this multidisciplinary model for collaborative learning opportunities could be a potentially good tool in the current postgraduate medical education in Pakistan. However, this pilot study will probably best be used to answer different research questions from the current written framework. I recommend that this manuscript be revised to answer different research questions, such as the strengths of the methods, feasibility, duration, cost, and possible adverse aspects of the approach they used in this multidisciplinary multi-institute online course.

Below is my list of some possible improvements for this manuscript:

1. The authors may want to reframe the whole manuscript to think about what the main points they would like to tell the readers about this pilot study – the strengths of this ‘novel’ approach, how feasible is this online course especially for this highly-stressed PG population, or what the challenges they expected when carrying out this kind of approach in a larger scale. It would be helpful for the audience to know how this approach is ‘novel’ in terms of a virtual learning environment (VLE), particularly in this post-COVID era, online learning has become the norm of learning mode globally. It would be informative to know the current or standard learning setting in Pakistan.

2. The results of the focus group discussion on needs of metabolic bone disorders online course were informative of the specific design of the actual online course: for example, total time commitment is about 1-2 hours per week, laptop over cellphone, and downloadable presentations. The reflection of the future inclusion of dentistry is interesting and worth noticing. However, it was not clear which parts of this online course reflected the strengths of a multidisciplinary multi-institute approach. If the authors can discuss which contribution each discipline has made in terms of the content and teaching tools, it would be more convincing for the readers to see how this approach is helpful.

3. The course execution, evaluation and feedback in the results section did not provide much information. One of the main advantages of qualitative studies is that the researchers could understand the participants in a much deeper way. Overall, not much information of the quality, feasibility, and other suggestion for this online course were obtained in this qualitative study. What were the actual patterns of the time the individual participants spent on each activity of this online course? Almost all the PGs were working long hours and did any of the participants rush through this online course? Why did no participant pass the end of the module assessment in one attempt? Was that expected? Why did only 47% of the facilitators provide their feedback of virtual teaching experience in the online survey? There is a lot more a qualitative study can provide, and I encourage the authors to provide more information to support their conclusion that ‘an institute with adequate facilities for faculty training can develop online multidisciplinary modules to benefit institutes in resource limited settings.’

4. Some other questions / comments I have here: What capacity has this online course built on the facilitators in this research team? Did the researchers encounter any challenge in the developing and executing process of this online course? On line 59, average hospital remain is probably better understood as hospital stay of 7-10 days. What does BCQs mean in line 263? Is it a typo for MCQs? Based on the PLOS Data Policy, it is required for the authors to make all data underlying the findings fully available. The authors may need to consider making their data available. Otherwise, it may not be possible to be published in PLOS.

Reviewer #3: The manuscript is an important addition to broader discourse on virtual learning and its need in a post covid era. However, some major revisions are needed to make it more concise and comprehendible.

1- the introduction to study lacks appropriate literature review. Add studies on previous findings about the impacts of online resources in medical education. Different perspectives should be explored about importance of online modules and their usefulness in developed vs developing countries.

2- the problem statement is not very clear. the justification of study does not align with problem statement. Does the study aim to produce the online module for outreach and facilitation in events of disruption (such as covid) or the online module is aimed to provide better course resources then currently available course resources?

3- Line 66-68: the argument is flawed. Does the status of city hinder the outreach of study material? provide the evidence.

4-Line 78-80: "The Aga Khan University’s ethical review committee’s acceptance (ERC number 5415-Pat-ERC-18) was taken, and faculty and PGs provided informed consent before the pilot was conducted" but the authors does not provide any ethics statement with their manuscript.

5-The sampling framework is not explained in detail. It would be useful for readers understanding, if a flow chart or table if formulated including the number of participants involved at stage (e.g. number of PGs for FGDs, number of faculty, number of PGs for MOOC testing).

6- What criteria were used to select PGs and faculty? How many invitations were sent and what were response rate?

7- The PGs involved in FGDs were different from PGs involved in execution of course or same? Clarify in manuscript.

8- The faculty are an important stakeholder. Why the FGDs does not involve any faculty?

9- Line 98: provide full form of MBD.

10- Line 106: "with risk factors and signs and symptoms" the use of and is made twice, which should be corrected. there are several other places where same mistake is made. Re-check the grammar and spelling carefully.

11-Line 116: Who are the facilitators in Phase IV-Course execution? explain.

12-Line 130: Provide full form of EMQ. what is indicated by "below level". Provide concrete definition of below level

13-Line 143: "The qualitative analytical method proposed by Braun and Clarke was used to conduct thematic

144 analysis of the FGD following the six phases." Provide in-text citation of Braun and Clarke's work.

14-Table 2b: "Understand the prevalence of osteoporosis in by age and gender in Pakistan" should be "Understand the prevalence of osteoporosis by age and gender in Pakistan". also check the rest of document for clarity and language comprehension.

15-Table 2b: Provide full form of FLS, BDM and FLAX. Check the whole document for full explanation of any abbreviation at its first use.

16-Line 201: Section Results-Course execution. "Ten PG trainees were enrolled in the course", but the figure 4 and 5a provides post-test results and evaluation of nine participants (n=9). Did all PG trainees not participated in post-test and evaluation? Explain

17-the discussion is more focused on course content rather then usefulness of involving PGs in course formulation process, the outreach impacts of courses, the constraints or problems faced in designing online courses and faculty training. these should be added in discussion. The discussion about course content should be added in literature review or methods sections to justify the course outline.

18-The study should add future plans in conclusion that how this MOOC will be shared to a wider medical audience.

Reviewer #4: Please review the attached file's comment boxes for required considerations. There are some minor corrections required regarding English formatting and grammar. However the methodology section lack clarification on selection criteria of the experts... ( may include specialty, experience of teaching, experience in online teaching etc etc). Result section shows nicely presented data , what I would like to know whether pre-post test result difference was significant or not. The course has been evaluated to level 2, study limitations or future directions can talk about its need to evaluate on further higher levels.

6. PLOS authors have the option to publish the peer review history of their article (what does this mean?). If published, this will include your full peer review and any attached files.

Reviewer #1: **Yes: **Sundus Tariq

Reviewer #2: **Yes: **Mei-kuang Chen

Reviewer #3: No

Reviewer #4: No

---

## [Author Response · Author response to Decision Letter 0]

7 Jul 2023

To the Editor PLOS ONE

Dated: 16/06/2023

Subject: Authors’ Responses to Reviewers’ Comments for Article Titled ‘Developing and piloting an online course on osteoporosis using a multidisciplinary multi-institute approach- a cross-sectional qualitative study’

Dear Editor, we have addressed all the comments and suggestions made by the reviewers point by point below. Additionally, all recommended modifications are made in track changes in the manuscript. 

1. Is the manuscript technically sound, and do the data support the conclusions? The manuscript must describe a technically sound piece of scientific research with data that supports the conclusions. Experiments must have been conducted rigorously, with appropriate controls, replication, and sample sizes. The conclusions must be drawn appropriately based on the data presented.

Reviewer #1: Yes, Reviewer #2: Partly, Reviewer #3: Partly, Reviewer #4: Partly

Reply: Not applicable (N/A)

2. Has the statistical analysis been performed appropriately and rigorously?

Reviewer #1: Yes, Reviewer #2: Yes, Reviewer #3: N/A, Reviewer #4: N/A

Reply: N/A

3. Have the authors made all data underlying the findings in their manuscript fully available? 

Reviewer #1: No, Reviewer #2: No, Reviewer #3: Yes, Reviewer #4: Yes

Reply: Thank you for your comment. The data has been uploaded on a repository and URL is shared in the editorial system. We have also made efforts to present and describe the complete data in the form of flow charts, tables, and figures within the results.

4. Is the manuscript presented in an intelligible fashion and written in standard English? PLOS ONE does not copyedit accepted manuscripts, so the language in submitted articles must be clear, correct, and unambiguous. Any typographical or grammatical errors should be corrected at revision, so please note any specific errors here.

Reviewer #1: Yes, Reviewer #2: Yes, Reviewer #3: No Reviewer #4: Yes

Reply: Thank you for your comment. We have tried to make the manuscript scientifically and intellectually coherent. Modifications are in track changes. 

5. Reviewer #1: Thank you for giving me the opportunity to review this well perceived and well written article titled, "Developing and piloting an online course on osteoporosis using a multidisciplinary multi-institute approach- a cross-sectional qualitative study". The methodology adopted is elaborated in detail and is reproducible. Moreover there is excellent flow of concepts. Only the references that are used are old. The authors need to discuss the results in light of recent literature preferably from within the last 5 years.

Reply: Thank you for your comments. We have revised the references in track changes to reflect current literature.

6. Reviewer #2: Paper Review: Developing and piloting an online course on osteoporosis using a multidisciplinary multi-institute approach- a cross-sectional qualitative study. This paper was about a cross-sectional qualitative study on a pilot online course of osteoporosis using a multidisciplinary multi-institute approach. It should be helpful for the postgraduates (PGs) in the medical education system to have the opportunities to learn from a virtual collaborative environment. This study showed that it was possible to design a multidisciplinary online course on osteoporosis, but the evidence was rather weak in terms of the educational impact of a novel educational tool described in this manuscript. The participating PGs were nominated by their faculty- the sample probably was not representative of the PG population in the country of study. Also, the qualitative nature of this study was not the best approach to examine the educational impact of this ‘novel’ tool. The main research question depicted in the abstract seemed to be ‘what was the educational impact of this novel educational tool on teaching and learning capabilities of PGs and course facilitators?”, and therefore the results only focused on the ‘quantitative’ findings based on the different measures related to the online course. Overall, this multidisciplinary model for collaborative learning opportunities could be a potentially good tool in the current postgraduate medical education in Pakistan. However, this pilot study will probably best be used to answer different research questions from the current written framework. I recommend that this manuscript be revised to answer different research questions, such as the strengths of the methods, feasibility, duration, cost, and possible adverse aspects of the approach they used in this multidisciplinary multi-institute online course. Below is my list of some possible improvements for this manuscript: 1. The authors may want to reframe the whole manuscript to think about what the main points they would like to tell the readers about this pilot study – the strengths of this ‘novel’ approach, how feasible is this online course especially for this highly-stressed PG population, or what the challenges they expected when carrying out this kind of approach in a larger scale. It would be helpful for the audience to know how this approach is ‘novel’ in terms of a virtual learning environment (VLE), particularly in this post-COVID era, online learning has become the norm of learning mode globally. It would be informative to know the current or standard learning setting in Pakistan.

Reply: We have discussed this in the introduction that why VLE is a novel concept in Pakistan. Please note in Pakistan, postgraduate medical education follows a structured framework. It involves residency training in accredited institutions, with defined curricula and syllabi for each specialty. Doctors gain hands-on experience through clinical rotations and have opportunities to attend workshops and seminars for continuous learning and are encouraged to engage in research. Assessments and examinations are conducted to monitor progress and obtain postgraduate degrees or certifications. Virtual learning environments represent a novel concept in this aspect, as they introduce technology-enabled platforms that overcome geographical barriers, enhance interactivity and collaboration, support continuous professional development, and adapt to changing circumstances, ultimately revolutionizing the way postgraduate medical education is delivered and experienced. 

While we recognize that the current evidence may have certain limitations, we believe that our findings contribute valuable insights to the field of education. The study represents an initial exploration of the novel educational tool and lays the groundwork for future research and more extensive investigations. The manuscript has been restructured to reflect the highlights of the study. 

7. The results of the focus group discussion on needs of metabolic bone disorders online course were informative of the specific design of the actual online course: for example, total time commitment is about 1-2 hours per week, laptop over cellphone, and downloadable presentations. The reflection of the future inclusion of dentistry is interesting and worth noticing. However, it was not clear which parts of this online course reflected the strengths of a multidisciplinary multi-institute approach. If the authors can discuss which contribution each discipline has made in terms of the content and teaching tools, it would be more convincing for the readers to see how this approach is helpful.

Reply: Thank you for your positive comments. Thank you for your valuable comment regarding the clarity of the strengths associated with the multidisciplinary multi-institute approach in our online course. We appreciate your suggestion to provide a more explicit discussion on the contributions of each discipline in terms of content and teaching tools. We have revised the manuscript to provide a detailed explanation of how the multidisciplinary approach is integrated into the course. We now explicitly discuss the specific contributions of each discipline, highlighting their expertise and unique perspectives that enrich the content and teaching tools.

8. The course execution, evaluation and feedback in the results section did not provide much information. One of the main advantages of qualitative studies is that the researchers could understand the participants in a much deeper way. Overall, not much information of the quality, feasibility, and other suggestion for this online course were obtained in this qualitative study. What were the actual patterns of the time the individual participants spent on each activity of this online course? Almost all the PGs were working long hours and did any of the participants rush through this online course? Why did no participant pass the end of the module assessment in one attempt? Was that expected? Why did only 47% of the facilitators provide their feedback of virtual teaching experience in the online survey? There is a lot more a qualitative study can provide, and I encourage the authors to provide more information to support their conclusion that ‘an institute with adequate facilities for faculty training can develop online multidisciplinary modules to benefit institutes in resource limited settings.’

Reply: The course execution, evaluation and feedback in the results section have been improved and sections have been expanded to provide more elaborate clarification of the outlined concerns. All edits are in track changes.

9. Some other questions / comments I have here: What capacity has this online course built on the facilitators in this research team? Did the researchers encounter any challenge in the developing and executing process of this online course? On line 59, average hospital remain is probably better understood as hospital stay of 7-10 days. What does BCQs mean in line 263? Is it a typo for MCQs? Based on the PLOS Data Policy, it is required for the authors to make all data underlying the findings fully available. The authors may need to consider making their data available. Otherwise, it may not be possible to be published in PLOS.

Reply: These concerns have been addressed in Phase II and Phase IV of the results, i.e. faculty development and course execution. We have added how the facilitators were trained on developing and running a course on VLE and they further trained more facilitators, and a ripple effect was created. Feedback gathered from facilitators (Results and also in figure 5b) describes the challenges faced in development and course execution. Line 59 has been corrected. BCQs have been elaborated. Data has been submitted as per PLOS Data Policy.

10. Reviewer #3: The manuscript is an important addition to broader discourse on virtual learning and its need in a post covid era. However, some major revisions are needed to make it more concise and comprehendible. The introduction to study lacks appropriate literature review. Add studies on previous findings about the impacts of online resources in medical education. Different perspectives should be explored about importance of online modules and their usefulness in developed vs developing countries.

Reply: Your comment has been noted. We have made the required modifications in tracked changes in the Introduction section now. New literature has been added in track changes. 

11. the problem statement is not very clear. the justification of study does not align with problem statement. Does the study aim to produce the online module for outreach and facilitation in events of disruption (such as covid) or the online module is aimed to provide better course resources then currently available course resources?

Reply: Problem statement has been revised in introduction. The online course aims to provide standardization and uniformity of treatment of osteoporosis across Pakistan. PG trainees can take it at their own pace and availability. We have tried to improve the problem statement in the last para of the introduction section. This course was developed before COVID, and the aim was not designed to keep disruptive events in mind. However, it can be useful in both circumstances.

12. Line 66-68: the argument is flawed. Does the status of city hinder the outreach of study material? provide the evidence.

Reply: Information acquired through focus group discussions showed that the PG trainees across Pakistan did not have good command over the concepts regarding management and treatment of osteoporosis. We have rephrased the sentence to reflect this.

13. Line 78-80: "The Aga Khan University’s ethical review committee’s acceptance (ERC number 5415-Pat-ERC-18) was taken, and faculty and PGs provided informed consent before the pilot was conducted" but the authors does not provide any ethics statement with their manuscript.

Reply: Ethics statement has been added in Methods. 

14. The sampling framework is not explained in detail. It would be useful for readers understanding, if a flow chart or table if formulated including the number of participants involved at stage (e.g. number of PGs for FGDs, number of faculty, number of PGs for MOOC testing).

Reply: Please see Fig 2 for the study workflow.

15. What criteria were used to select PGs and faculty? How many invitations were sent and what were response rate?

Reply: The selection criteria for both have been added in the methods section.

16. The PGs involved in FGDs were different from PGs involved in execution of course or same? Clarify in manuscript.

Reply: We have clarified this in the methods.

17. The faculty are an important stakeholder. Why the FGDs does not involve any faculty?

Reply: The decision was made to exclude faculty perceptions from the focus group discussion for postgraduate trainees to create a safe and unbiased environment where trainees could freely express their opinions and experiences. This allowed for candid insights solely based on trainees' perspectives, without potential influence or bias from faculty members. 

18. Line 98: provide full form of MBD.

Reply: It has been provided in the Introduction section now.

19. Line 106: "with risk factors and signs and symptoms" the use of and is made twice, which should be corrected. there are several other places where same mistake is made. Re-check the grammar and spelling carefully.

Reply: Thank you for your comment. The grammar has been checked and corrected throughout the manuscript.

20. Line 116: Who are the facilitators in Phase IV-Course execution? Explain.

Reply: The section now provides a clear explanation of the facilitators.

21. Line 130: Provide full form of EMQ. what is indicated by "below level". Provide concrete definition of below level

Reply: EMQ refers to “extended matching questions”. Below level means he/she is unable to pass the module and this has been explained in the manuscript.

22. Line 143: "The qualitative analytical method proposed by Braun and Clarke was used to conduct thematic 144 analysis of the FGD following the six phases." Provide in-text citation of Braun and Clarke's work.

Reply: The in-text citation has been provided.

23. Table 2b: "Understand the prevalence of osteoporosis in by age and gender in Pakistan" should be "Understand the prevalence of osteoporosis by age and gender in Pakistan". also check the rest of document for clarity and language comprehension.

Reply: The manuscript has been checked and improved for clarity and language comprehension.

24. Table 2b: Provide full form of FLS, BDM and FLAX. Check the whole document for full explanation of any abbreviation at its first use.

Reply: The abbreviations have been adequately explained throughout the manuscript. 

25. Line 201: Section Results-Course execution. "Ten PG trainees were enrolled in the course", but the figure 4 and 5a provides post-test results and evaluation of nine participants (n=9). Did all PG trainees not participated in post-test and evaluation? Explain

Reply: The correct number is nine, and has been corrected throughout the manuscript.

26. the discussion is more focused on course content rather then usefulness of involving PGs in course formulation process, the outreach impacts of courses, the constraints or problems faced in designing online courses and faculty training. these should be added in discussion. The discussion about course content should be added in literature review or methods sections to justify the course outline.

Reply: These concerns have been appropriately addressed in the relevant sections now. 

27. The study should add future plans in conclusion that how this MOOC will be shared to a wider medical audience.

Reply: Conclusion has been revised and way forward has been discussed as per your suggestions.

28. Reviewer #4: Please review the attached file's comment boxes for required considerations. There are some minor corrections required regarding English formatting and grammar. However the methodology section lack clarification on selection criteria of the experts... ( may include specialty, experience of teaching, experience in online teaching etc etc). Result section shows nicely presented data , what I would like to know whether pre-post test result difference was significant or not. The course has been evaluated to level 2, study limitations or future directions can talk about its need to evaluate on further higher levels.

Reply: Thank you for your comments. The suggested changes have been addressed and highlighted in track changes throughout the document. 

Sincerely

Lena Jafri

---

## [Decision Letter · Decision Letter 1]

10 Aug 2023

PONE-D-22-28614R1Developing and piloting an online course on osteoporosis using a multidisciplinary multi-institute approach- a cross-sectional qualitative studyPLOS ONE

Dear Dr. Jafri,

Thank you for submitting your manuscript to PLOS ONE. After careful consideration, we feel that it has merit but does not fully meet PLOS ONE’s publication criteria as it currently stands. Therefore, we invite you to submit a revised version of the manuscript that addresses the points raised during the review process.

ACADEMIC EDITOR: Please check reviewer 2 comments.Please ensure that your decision is justified on PLOS ONE’s publication criteria and not, for example, on novelty or perceived impact.

We look forward to receiving your revised manuscript.

Kind regards,

Mukhtiar Baig, Ph.D.

Academic Editor

PLOS ONE

Journal Requirements:

Reviewers' comments:

Reviewer's Responses to Questions

**Comments to the Author**

1. If the authors have adequately addressed your comments raised in a previous round of review and you feel that this manuscript is now acceptable for publication, you may indicate that here to bypass the “Comments to the Author” section, enter your conflict of interest statement in the “Confidential to Editor” section, and submit your "Accept" recommendation.

Reviewer #2: All comments have been addressed

Reviewer #3: (No Response)

2. Is the manuscript technically sound, and do the data support the conclusions?

Reviewer #2: Yes

Reviewer #3: Yes

3. Has the statistical analysis been performed appropriately and rigorously? 

Reviewer #2: N/A

Reviewer #3: N/A

4. Have the authors made all data underlying the findings in their manuscript fully available?

Reviewer #2: Yes

Reviewer #3: Yes

5. Is the manuscript presented in an intelligible fashion and written in standard English?

Reviewer #2: Yes

Reviewer #3: Yes

6. Review Comments to the Author

Reviewer #2: Improvements:

1. The authors made all data underlying the findings in their manuscript fully available now.

2. I appreciate that the authors responded to my comments and revised their manuscript accordingly.

My only suggestion is that the abstract does not reflect the improvements the authors have done in the manuscript itself. Was the only goal of this study to gauge the educational impact of a novel educational tool? I don’t think that this manuscript has the sample to do this kind of evaluation in terms of educational impact. It does show the feasibility and potential of a powerful educational tool in the medical education in Pakistan. Also, the results and the conclusion in the abstract don’t seem to connect well. If the authors can summarize the strengths of qualitative part of their online course development process, the abstract will give a better overview of the approach they adopted in this manuscript.

There are a few minor points I want to address here:

1. This point may be irrelevant, and it can be ignored if the editors of this journal are fine with this. I just don’t understand why the authors responded ‘N/A’ to the Ethic Statement in their submission of this manuscript. This study did involve human participants. However, the authors did include the ethical statement in their revised manuscript. So it was taken care of in the methods section of the manuscript. Therefore, it can be an irrelevant point here.

2. Fig 5a and 5b were the count of each item in terms of course content and satisfaction of this innovative teaching and learning method? In the Phase V – course evaluation and feedback (starting from LINE 174 and ending in LINE 193 in the clean version of the revised manuscript), an online survey was on a Likert scale of 1-5. How do we interpret the fig 5a and 5b? Or are Fig 5a and 5b not related to this section?

3. In LINE 261, an extra T was put in with ‘Nine’ PG trainees in the clean version of the revised manuscript.

Reviewer #3: (No Response)

7. PLOS authors have the option to publish the peer review history of their article (what does this mean?). If published, this will include your full peer review and any attached files.

Reviewer #2: No

Reviewer #3: No

---

## [Author Response · Author response to Decision Letter 1]

25 Aug 2023

Authors’ Responses to Reviewers’ Comments

25/8/2023

Comments to the Author

1. If the authors have adequately addressed your comments raised in a previous round of review and you feel that this manuscript is now acceptable for publication, you may indicate that here to bypass the “Comments to the Author” section, enter your conflict of interest statement in the “Confidential to Editor” section, and submit your "Accept" recommendation.

Reviewer #2: All comments have been addressed

Reviewer #3: (No Response)

Authors’ Response: Thank you for your comments.

2. Is the manuscript technically sound, and do the data support the conclusions?

Reviewer #2: Yes

Reviewer #3: Yes

Authors’ Response: Thank you for your comments.

3. Has the statistical analysis been performed appropriately and rigorously?

Reviewer #2: N/A

Reviewer #3: N/A

Authors’ Response: Thank you for your comments.

4. Have the authors made all data underlying the findings in their manuscript fully available?

Reviewer #2: Yes

Reviewer #3: Yes

Authors’ Response: Thank you for your comments.

5. Is the manuscript presented in an intelligible fashion and written in standard English?

Reviewer #2: Yes

Reviewer #3: Yes

Authors’ Response: Thank you for your comments.

6. Review Comments to the Author

Reviewer #2: Improvements:

1. The authors made all data underlying the findings in their manuscript fully available now.

2. I appreciate that the authors responded to my comments and revised their manuscript accordingly.

My only suggestion is that the abstract does not reflect the improvements the authors have done in the manuscript itself. Was the only goal of this study to gauge the educational impact of a novel educational tool? I don’t think that this manuscript has the sample to do this kind of evaluation in terms of educational impact. It does show the feasibility and potential of a powerful educational tool in the medical education in Pakistan. Also, the results and the conclusion in the abstract don’t seem to connect well. If the authors can summarize the strengths of qualitative part of their online course development process, the abstract will give a better overview of the approach they adopted in this manuscript.

There are a few minor points I want to address here:

1. This point may be irrelevant, and it can be ignored if the editors of this journal are fine with this. I just don’t understand why the authors responded ‘N/A’ to the Ethic Statement in their submission of this manuscript. This study did involve human participants. However, the authors did include the ethical statement in their revised manuscript. So it was taken care of in the methods section of the manuscript. Therefore, it can be an irrelevant point here.

2. Fig 5a and 5b were the count of each item in terms of course content and satisfaction of this innovative teaching and learning method? In the Phase V – course evaluation and feedback (starting from LINE 174 and ending in LINE 193 in the clean version of the revised manuscript), an online survey was on a Likert scale of 1-5. How do we interpret the fig 5a and 5b? Or are Fig 5a and 5b not related to this section?

3. In LINE 261, an extra T was put in with ‘Nine’ PG trainees in the clean version of the revised manuscript.

Authors’ Response: 

Thank you for your comments and suggestions! We have reworded the abstract to reflect the aims and goals of the manuscript. With regards to point 1, we have included the ethical statement in the manuscript. Point 2 stands correct- a Likert scale was not used, and the explanation has been modified to reflect that. Point 3 has also been addressed.

Reviewer #3: (No Response)

---

## [Editor Report · Decision Letter 2]

4 Sep 2023

Developing and piloting an online course on osteoporosis using a multidisciplinary multi-institute approach- a cross-sectional qualitative study

PONE-D-22-28614R2

Dear Dr. Jafri,

We’re pleased to inform you that your manuscript has been judged scientifically suitable for publication and will be formally accepted for publication once it meets all outstanding technical requirements.

Kind regards,

Mukhtiar Baig, Ph.D.

Academic Editor

PLOS ONE

---

## [Editor Report · Acceptance letter]

12 Oct 2023

PONE-D-22-28614R2 

Developing and piloting an online course on osteoporosis using a multidisciplinary multi-institute approach- a cross-sectional qualitative study 

Dear Dr. Jafri:

I'm pleased to inform you that your manuscript has been deemed suitable for publication in PLOS ONE. Congratulations! Your manuscript is now with our production department. 

Kind regards, 

on behalf of

Professor Mukhtiar Baig 

Academic Editor

PLOS ONE